# Using TENS for Pain Control: Update on the State of the Evidence

**DOI:** 10.3390/medicina58101332

**Published:** 2022-09-22

**Authors:** Carol G. T. Vance, Dana L. Dailey, Ruth L. Chimenti, Barbara J. Van Gorp, Leslie J. Crofford, Kathleen A. Sluka

**Affiliations:** 1Department of Physical Therapy and Rehabilitation Science Department, Roy J and Lucille A Carver College of Medicine, The University of Iowa, Iowa City, IA 52242, USA; 2Department of Physical Therapy, St. Ambrose University, Davenport, IA 52803, USA; 3Division of Rheumatology & Immunology, Medical Center, Vanderbilt University, Nashville, TN 37232, USA

**Keywords:** pain, TENS, evidence

## Abstract

Transcutaneous electrical nerve stimulation (TENS) is a non-pharmacological intervention used in the treatment of acute and chronic pain conditions. The first clinical studies on TENS were published over 50 years ago, when effective parameters of stimulation were unclear and clinical trial design was in its infancy. Over the last two decades, a better understanding of the mechanisms underlying TENS efficacy has led to the development of an adequate dose and has improved outcome measure utilization. The continued uncertainty about the clinical efficacy of TENS to alleviate pain, despite years of research, is related to the quality of the clinical trials included in systematic reviews. This summary of the evidence includes only trials with pain as the primary outcome. The outcomes will be rated as positive (+), negative (−), undecided (U), or equivalent to other effective interventions (=). In comparison with our 2014 review, there appears to be improvement in adverse events and parameter reporting. Importantly, stimulation intensity has been documented as critical to therapeutic success. Examinations of the outcomes beyond resting pain, analgesic tolerance, and identification of TENS responders remain less studied areas of research. This literature review supports the conclusion that TENS may have efficacy for a variety of acute and chronic pain conditions, although the magnitude of the effect remains uncertain due to the low quality of existing literature. In order to provide information to individuals with pain and to clinicians treating those with pain, we suggest that resources for research should target larger, high-quality clinical trials including an adequate TENS dose and adequate timing of the outcome and should monitor risks of bias. Systematic reviews and meta-analyses should focus only on areas with sufficiently strong clinical trials that will result in adequate sample size.

## 1. Introduction

Transcutaneous electrical nerve stimulation (TENS) is a non-pharmacological intervention used in the treatment of acute and chronic pain conditions. TENS units are safe, inexpensive, over-the counter devices that deliver pulsed alternating current applied through electrodes placed on the skin [1,2]. The parameters of pulse frequency and pulse intensity are adjustable and linked to TENS efficacy [3]. TENS can be applied with low frequencies (LF), <10 Hz, high frequencies (HF), >100 Hz, or mixed frequencies (LF and HF). Frequencies of stimulation result in unique mechanisms of action for TENS, as outlined below. We summarize the mechanisms of action and factors that influence TENS efficacy, and describe and critique the use of TENS for pain control in a variety of patient populations. The findings of systematic reviews of TENS for pain management in the last nine years will be presented. We also highlight emerging evidence from randomized controlled trials (RCT) published in the last nine years that are not included in the SR. This article offers a concise review of the fundamental mechanisms of TENS as well as an up-to-date critique of current clinical research for TENS.

## 2. Mechanisms of TENS Underlying Analgesic Effects

TENS activates inhibitory mechanisms to reduce central excitability primarily in the central nervous system and to consequently reduce pain. TENS activates large afferent fibers in the periphery [4,5] that send input to the central nervous system. This is turn activates descending inhibitory systems that reduce hyperalgesia. Specifically, prior studies show that blockade of neuron activity in the spinal cord, rostral ventromedial medulla (RVM) and the periaqueductal gray (PAG) inhibit analgesic effects of TENS [6,7,8]. In parallel, studies in people with fibromyalgia (FM) show that TENS can restore central pain modulation, an indicator of central inhibition [9]. 

In healthy human volunteers, brain responses measured with electroencephalography (EEG), demonstrate reduced cortical activity with both LF and HF TENS. Reductions in activity of the primary somatosensory (S1) and motor (M1) cortices occurred with both frequencies; however, reduced connectivity between SI/M1 and the prefrontal cortex were found only with LF TENS [10]. Using functional magnetic resonance imaging of healthy volunteers, HF TENS reduced activity in the pons and connectivity between the pons and S1, while LF TENS increased functional connectivity in the medulla (RVM and subnucleus reticularis dorsalis) and decreased functional connectivity between the frontal cortices and the medulla [11]. In both of these studies, the intensity of HF TENS was strong but comfortable, while that of LF TENS was noxious. Thus, it is unclear if the differences in cortical activity responses between LF and HF TENS were due to frequency or intensity of stimulation.

## 3. Neurotransmitters and Receptors That Mediate TENS Analgesia

The analgesic effect of TENS is mediated through multiple receptors, but specific receptors utilized depend on TENS frequency. Early studies show that HF TENS (>50 Hz) increases the concentration of β-endorphins and methionine-enkephalin in the cerebrospinal fluid of human subjects [12,13] and that LF TENS (<10 Hz) analgesia was prevented by blockade of opioid receptors with naloxone [14]. In an animal model of inflammation, TENS produces analgesia that is prevented by blockade of opioid receptors in the RVM or spinal cord in a frequency-dependent manner [7,8]. HF TENS analgesia is blocked by δ-opioid receptor antagonists, and LF TENS is blocked by µ-opioid receptor antagonists in the spinal cord and RVM [7,8].This opioid-mediated analgesia produced by LF and HF TENS has been confirmed in human subjects [15,16,17]. Additionally, blockade of muscarinic M1 and M3 receptors and GABA_A_ receptors in the spinal cord reduces both LF and HF TENS analgesia [18,19]. LF TENS increases the release of serotonin, which produces analgesia through 5-HT2A and 5-HT3 serotonin receptors in the spinal cord [20]. The blockade of cannabinoid receptor 1 (CB1) in the PAG reversed both low- and high-frequency TENS analgesia and was associated with an increase in CB1 receptor expression [21]. Thus, TENS produces analgesia by activating endogenous inhibitory mechanisms in the central nervous system, involving opioid, GABA, serotonin, muscarinic, and cannabinoid receptors.

Repeated application of TENS, at the same dose (intensity, duration, and frequency) daily produces analgesic tolerance within several days in animals and in human studies [22,23]. There is cross tolerance at central opioid receptors such that δ-opioid agonists are not effective after repeated application of HF TENS and µ-opioid agonists are not effective after LF TENS application [22]. Similarly, in both human and animal subjects tolerant to exogenous µ-opioid agonists, LF TENS is ineffective while HF TENS analgesia remains effective [15,22]. Importantly, the tolerance can be alleviated with mixed-frequency TENS (LF and HF) or by increasing intensity by as little as 10% [24,25]. Furthermore, in our recent clinical trial, we showed no tolerance with repeated application of mixed frequency TENS, and in fact, we showed a cumulative effect over a month of treatment and continued effectiveness for 2 months [26]. Thus, mixed frequency TENS alleviates analgesic tolerance to maintain efficacy with repeated use. 

## 4. TENS Reduces Central Excitability

TENS directly reduces the activity of nociceptive dorsal horn neurons in uninjured animals and the sensitization of dorsal horn neuron in injured animals [27,28,29,30]. In parallel, there is a reduction in both primary and secondary hyperalgesia by both LF and HF TENS [31,32,33,34,35,36] and release of the excitatory neurotransmitters glutamate and substance P in the spinal cord dorsal horn in animals with inflammation, neuropathic, or incisional pain [37,38,39,40]. Furthermore, in people with FM and osteoarthritis, there is a reduction in pressure pain thresholds not only at the site of TENS stimulation but also at sites outside the area of application [9,41], implicating a reduction in central excitability. Both HF and LF TENS also reduces microglia and astrocyte activation in the spinal cord in both osteoarthritic and neuropathic pain animal models, and associated proinflammatory cytokines, intracellular messengers, and transcription factors [39,42,43]. The reduction in glutamate release and glial cell activation is blocked by opioid receptor antagonists [37,43]. Therefore, TENS reduces hyperalgesia through central mechanisms that increase inhibition and subsequently reduce central excitability. 

## 5. Peripheral Mechanisms of TENS

A few studies have shown peripheral effects of TENS. HF TENS reduces tissue-injury increases in substance P in dorsal root ganglia neurons [38], blockade of peripheral opioid receptors prevents the analgesia produced by LF and HF TENS [44,45,46], and blockade of CB1 receptors prevents production of analgesia by HF and LF TENS [21]. Thus, TENS may also alter excitability of peripheral nociceptors to reduce afferent input to the central nervous system.

Interestingly, in α-2a adrenergic knockout mice, both LF and HF TENS analgesia does not occur; this effect is mediated peripherally since the blockade of peripheral, but not spinal or supraspinal, α-2 receptors prevents the analgesia produced by TENS [33,47]. In humans, there are increases in blood flow with TENS at intensities greater than 25% above motor threshold that produce motor contractions [48,49,50,51]. Thus, some of the analgesic effects of TENS are mediated through peripheral adrenergic receptors modulating autonomic function.

## 6. Factors That Directly Affect TENS Efficacy

Our 2014 review of the evidence for TENS management emphasized the importance of factors influencing TENS efficacy including dosing parameters, timing of the outcome measure, interactions with medications, and repeated use [52]. In the current review of the literature common problems in determining if an effective dose or appropriate outcome measure was utilized are still evident. Overall, systematic reviews commonly describe that there is limited reporting on TENS parameters, adverse events, high risk of bias, low number of participants, inadequate blinding, and limited reporting of methodology [53,54,55,56,57,58,59,60,61,62,63]. This has led to a number of systematic reviews showing inconclusive or weak evidence. It is routinely recommended that future RCTs be well-designed to reach robust conclusions.

The continued uncertainty about the clinical efficacy of TENS to alleviate pain, despite years of research, is related to the quality of the clinical trials included in systematic reviews [3,56,64]. Clinical practice guidelines are mixed with some recommending TENS as an adjunct treatment for osteoarthritis or rheumatoid arthritis [65], for adults with hip fracture when pain is not managed with other interventions [66], as an adjunct intervention to an exercise program for chronic cervical–thoracic pain [67], and for chronic low back pain as an adjunct to higher order interventions (exercise and NSAIDS) [68]. Another clinical practice guideline does not recommend TENS for osteoarthritis [69]. For other types of pain, such as acute and subacute low back pain, knee osteoarthritis, and post operative total knee arthroplasty, there are recommendations against TENS use [68,69,70,71]. These guidelines are mainly based on published systematic reviews that do not account for differences in effect size due to dosing. Nearly a decade ago, our group suggested that factors related to dosing of TENS such as adequate stimulation parameters and timing of outcome measurement were critical to showing efficacy of TENS and thus should be included in systematic reviews. In 2020, we published the results of a large, randomized controlled clinical trial, using a strong sample size (*n* = 301), adequate parameters (mixed frequency, and strong but comfortable intensity), and adequate timing of outcome measure (testing during TENS active period after 1 month of use) and showed efficacy of TENS on movement-evoked pain in women with FM [26]. Similarly, a recent meta-analysis showed that TENS at strong but comfortable intensity, with the assessment of pain measured during or immediately after TENS showed a significant and large effect size of TENS when compared with placebo (SMD = 0.96) [72]. 

Our recent RCT also showed that TENS was safe, with no serious adverse events reported in over 300 individuals, and the number needed to treat was between 3 and 4 for individuals with FM [2]. These data suggest that the benefit is similar to pharmaceutical treatments currently used for chronic pain [73,74,75] and that TENS has a lower risk because it substantially safer than pharmaceutical treatments [74,76,77,78]. Furthermore, TENS is cost-effective and is available over the counter in many countries and thus can be used as part of a self-management strategy similar to heat, cold, and acetaminophen or NSAIDs. As with most treatments, TENS may be effective for a subpopulation of individuals. Again, as an example, our recent RCT on TENS in FM showed that 44% of individuals had a clinically meaningful reduction in pain with TENS [2,26]. This percentage of responders to treatment is again similar to that observed for pharmaceutical agents for pain [79]. Thus, uncertainty about the clinical efficacy of TENS is primarily due to limitations of existing literature, as Johnson and colleagues [64] highlighted in terms of the sample sizes from RCTs included in meta-analysis and the risk for bias and heterogeneity of the systematic reviews with the pooled analysis. 

Keeping in mind the strengths and weaknesses of systematic reviews, we summarize the evidence by including only trials with pain as the primary outcome, rating the outcome as positive (+), negative (−), undecided (U), or equivalent to other effective interventions (=). For the purpose of our review, the descriptor of “undecided” will be used to classify trials where results were described as uncertain, inconclusive, undetermined, weak or conflicting by the authors. Table 1 provides a summary review of the systematic reviews and meta-analyses discussed below. Table 2 outlines the pain outcome measures, adverse event reporting, TENS ratings and TENS recommendations for the same articles included in this review.

## 7. Evidence of TENS for Pain Management 

### 7.1. Acute Pain 

For TENS used in individuals with acute pain, systematic reviews in the last nine years have focused on four areas with respect to acute pain: acute pain in general [1], the pre-hospital setting [81], paramedic pain management of femur fracture [80], and acute low back pain [53]. 

A Cochrane review of TENS use for acute pain [1], last updated in 2015, included a total of 19 clinical trials with pooled analysis of 1346 individuals. Importantly, this systematic review only included trials where TENS was delivered as a stand-alone treatment at an adequate dose emphasizing a strong but comfortable patient sensation. The types of acute pain were varied and categorized as procedural pain or non-procedural pain. Six trials were added to the data from the previous systematic review in 2011. Overall, in these six clinical trials, the authors found that active TENS was better than placebo TENS; however, the instructions and timing for reporting pain were not consistent among the studies. Pain reduction was rated by visual analog scale (VAS), numerical rating scales (NRS); verbal rating scales (VRS), or McGill Pain Questionnaire (MPQ). This review included the most appropriate and descriptive accounting of the TENS dose in the inclusion criteria for trials and collected adverse events (AE). Trials were included with placebo TENS, no treatment, and pharmacological and non-pharmacological interventions as comparators with active TENS. The success of the TENS depended on the application of active TENS and varied between the clinical trials. Treatment parameters were incomplete for replication of the studies. Minor adverse events reported included itching and redness at the site of TENS stimulation, or a dislike of the sensation of TENS. Active TENS may reduce the intensity of acute pain; however, the evidence was classified as tentative.

A confounder persists when evaluating TENS for analgesia in systematic reviews and occurs when multiple interventions are included to address the pain condition. For example, paramedic pain management of femur fracture included a variety of interventions, with the inclusion of only one trial examining the use of TENS. For TENS use with femur fracture [80], 72 participants received either active TENS (100 Hz, pulse width: 200 μs, voltage: 2 mA, time: 30 min) or sham TENS. Pain in the active TENS group decreased from 89 ± 9 mm to 59 ± 6 mm, whereas the sham group decreased from 86 ± 12 mm to 79 ± 11 mm (VAS, 100 mm scale).

In the pre-hospital setting [81], active TENS compared with sham TENS demonstrated clinically meaningful reductions in pain severity (VAS, 100 mm scale) and anxiety associated with pain. The review included four RCTs with pooled analysis of 261 patients. The TENS parameters were 100 Hz and 2 mA, with a comparison of active TENS compared with sham TENS. The mean reduction in the pooled analysis was 30 mm (95% CI 21–4; P < 0.0001), with a range of 33–55 mm. In an RCT, postoperative pain was assessed in 78 adults undergoing cholecystectomy [102]. Pain was assessed before and after a 30 min TENS (150 Hz, 75 µs) treatment delivered at the maximally tolerated intensity without causing muscle contraction or noxious stimulation. Using the criterion of two points or greater decrease in pain on a 0–10 scale, active TENS decreased pain 53.8% of the time as compared with placebo TENS at 11.5%. 

For acute low back pain [53], a systematic review included three placebo-controlled studies with a pooled analysis of 192 participants [53]. In one study, a 30 min TENS treatment while in transport to the hospital reduced pain by 28.0 mm (95% CI: −32.7 to −23.3) when compared with the placebo. However, the remaining two studies where TENS was utilized over 4–5 weeks in standard setting or prior to an exercise program for 4 weeks, were inconclusive. For all three studies, there was limited reporting on TENS parameters, adverse events, or follow-up beyond the acute care setting. The evidence overall was inconclusive with respect to TENS and acute low back pain primarily due to the low quality of the evidence. A clinical practice guideline (CPG) for older adults with hip fracture supports the use of TENS when pain is unmanaged with usual interventions [66].

Thus, the majority of systematic reviews support the use of TENS for acute pain. Importantly, in reviews where appropriate dosing of TENS was an inclusion criterion or was reported, there were clinically meaningful reductions in pain. While there are still weaknesses with many studies in terms of sample size, dosing parameters, or timing of outcomes, there is a growing body of literature supporting the use of TENS for acute pain over a variety of conditions. 

### 7.2. Chronic Pain

A Cochrane review for TENS in chronic pain included systematic review of systematic reviews for chronic pain [87], excluding headache or migraines. The review included nine systematic reviews with 51 clinical trials comparing active TENS to sham TENS with a pool of 2895 participants. The authors sought to collect adverse events as a primary outcome; however, they were unable due to inconsistent reporting of adverse events. In the systematic review, the quality of the clinical trials was rated as very low due to the risk of bias, small sample sizes, and limitations in methodology. Ultimately, the evidence was determined to be uncertain for TENS efficacy; however, this finding is potentially misleading because adequate TENS dose was not mentioned as a metric for inclusion of trials. The comparisons of interest included in the review included TENS versus usual care or no treatment/waiting list control, TENS versus sham TENS, TENS plus active intervention versus active intervention alone, and different types of active TENS with varying parameters, which adds to the uncertainty of the findings.

### 7.3. Fibromyalgia

For adults with FM, three systematic reviews were identified [57,88,89]. In all three systematic reviews, the primary outcome was pain using the VAS scale. Secondary outcomes included quality of life, fatigue, analgesic consumption, and sleep.

A Cochrane systematic review for TENS in adults with FM identified eight studies with a pool of 315 adults (94% women) [57]. With respect to the pain measures, only one of eight studies used a 30% or greater pain reduction compared with another treatment. Additional outcomes included patient global impression of change, and adverse events. A variety of comparison groups were utilized across the eight studies (TENS compared with the placebo, TENS compared with no treatment, and TENS compared with exercise alone). The evidence was inconclusive due to a small number of studies, limited reporting of methods, and limited detail regarding TENS application and parameters. The authors had planned to conduct a sub-analysis comparing the trials use of adequate stimulation intensity to those who described the stimulation to be “faint”, however, the low quality of evidence and limited detail impeded the completion of the sub-analysis.

A 2017 meta-analysis by Salazar et al. included nine studies including 301 participants with FM [89]. The meta-analysis concluded a positive effect of active electrical stimulation treatment versus placebo (−1.24 (95% CI: −2.39 to −0.08; I2: 87%, *p* = 0.04)). The trials reviewed demonstrated a risk of bias, limitations in methodology, and a small number of studies. TENS demonstrated low-quality evidence, with concerns around blinding and allocation.

Another review article included 11 studies regarding the effects of a variety of physical agents with only one TENS specific trial [88]. The TENS trial (*n* = 36) reported pain reduction (VAS 10 cm) for three groups: single active TENS, dual active TENS (200 μs, 2 and 100 Hz, 60 mA, 20 min: twice a day for 7 days) and placebo TENS. Pain reduction for the dual TENS group was 4.0 cm (*p* < 0.02); that for the single TENS group was 2.5 cm (*p* < 0.05); and for the placebo group, there was no pain reduction. [103] FM is difficult to treat so the effort to include many interventions in a review is understandable; however, this strategy does not assist to demonstrate the specific efficacy of TENS for FM.

To address weaknesses in prior studies, a recent RCT [26] for women with FM (*n* = 301) compared active-TENS (*n* = 101) with placebo-TENS (*n* = 99) or no-TENS (*n* = 99). This study used a mixed frequency TENS to prevent tolerance, applied TENS at a strong but comfortable intensity, and examined effects on movement and resting pain during TENS. There was a significant difference in movement-evoked pain after 4 weeks of home use in the active TENS compared with the placebo TENS (group mean difference –1.0 [95% confidence interval: –1.8 to –0.2]; *p* = 0.008; NRS, 0 to 10) or the no TENS group (group mean difference –1.8 [95% confidence interval: –2.6 to –1.0]; *p* < 0.001). A reduction in movement-evoked fatigue was also reported in the active TENS group versus the placebo TENS group (group mean difference –1.4 [95% confidence interval: –2.4 to –0.4]; *p* = 0.001) and versus the no TENS group (group mean difference –1.9 [95% confidence interval: –2.9 to –0.9]; *p* ≤ 0.0001). In a secondary analysis of this data [2], the authors identified TENS responders (30% reduction in pain or 20% reduction in fatigue) and used logistic regression analyses to identify factors that predict a positive outcome with TENS use. The single best predictor for a clinically meaningful reduction in pain was the change in movement evoked pain (six-minute walk test) from before and during the initial 30 min TENS treatment [2]. An additional predictor of pain responders was less FM disease severity. There were no serious adverse events and minimal minor adverse events, which included the following: nausea or pain with TENS, anxiety, skin irritation and itchiness, with all of these events being manageable by a clinician. Number needed to harm (NNH) ranged between 20 and 100 for minor adverse events. Number needed to treat (NNT) to obtain one additional pain responder was 3.3. This study serves as an example of a well-designed, adequately powered trial that shows a clinically meaningful reduction in pain with TENS. Furthermore, it shows that TENS is safe with no serious adverse events, minor modifiable adverse events, and a large NNH. 

### 7.4. Knee Osteoarthritis

The 2019 American College of Rheumatology/Arthritis Foundation (ACR) guidelines for management of OA of the knee strongly recommends against the use of TENS for OA pain [69], and TENS is absent from the APTA CPG for total knee arthroplasty (TKA) [71]. These guidelines are based on RCTs, systematic review and meta-analyses; however, the ACR 2019 guideline recommendation is made without the inclusion of TENS manuscripts in the results or in the discussion of summary of papers excluded from the review. The APTA CPG included non-pharmacological interventions in the PICO question and specifically included TENS in the search terms; however, this guideline also does not include manuscripts addressing TENS utilization for postoperative TKA. 

In contrast, several systematic reviews show that the evidence supporting the short-term effectiveness of TENS for relief of knee osteoarthritis (OA) pain is moderately strong [90,91,92,93,104]. HF TENS provides a statistically significant level of pain relief in patients with chronic knee OA pain, yet the standardized mean difference compared with a control group may not be of a clinically meaningful magnitude [90,93]. Of note, the effect size of HF TENS (−16.63, 95% CI: −24.6 to −8.7) is larger than the effect size of medication (−7.1, 95% CI: −12.1 to −2.2) [90] Active-TENS also provided greater pain relief than placebo-TENS for post-operative total knee arthroplasty pain, although the standardized mean difference was less than 1 point on a 0-to-10-point scale [91,92]. While the difference may not be clinically meaningful, it also affects other aspects of healthcare, such as a statistically significant lower amount of post-operative opioid consumption [91,92], which could contribute to long-term health consequences. Different types of electrical stimulation may contribute to some of the variability in outcomes between studies, with interferential current and HF TENS having larger effect sizes than other forms of stimulation, such as LF TENS [95]. Due to a lack of studies, meta-analyses were unable to draw conclusions about long-term effects of TENS for knee OA pain. TENS use was not supported by a RCT claiming TENS should not be further researched for management of pain in patients with knee OA [58]. Though this trial included a power analysis and sufficient numbers per arm, the TENS intervention may not have utilized an adequate dose to achieve a significant reduction in pain. This is currently an emerging area of research with at least one RCT supporting that TENS can maintain effectiveness at 1 year [94]. 

### 7.5. Musculoskeletal Conditions

The evidence for TENS when treating various musculoskeletal conditions remains mostly uncertain. In addition to the routinely reported problems such as low quality of trials, lack of sufficient studies or sample size, lacking clear methodology, and limited detail of parameters of application, most systematic reviews include TENS as one of many interventions and in the trials included, TENS is often compared with numerous other interventions (exercise, other electrotherapies, ultrasound, and injections) rather than placebo or sham TENS, or standard care. 

Seven RCTs representing 651 participants with chronic neck pain were included in a systematic review where the effect of TENS was uncertain due to insufficient evidence. These trials could not be combined in meta-analysis [59]. Although TENS parameters were listed for the trial, duration was noted to be between 15 and 60 min, with the number of sessions ranging from 1 to 60. Intensity was described as tolerable tingling without muscle contraction. The authors recommend that future RCTs be well-designed to reach robust conclusion and should compare conventional TENS vs. sham, utilizing the IMMPACT (Initiative of Methods, Measurement and Pain Assessment in Clinical Trials) when planning the selection and measurement of outcomes for future studies. The American College of Occupational and Environmental Medicine 2018 CPG suggest that while TENS is not recommended for acute/sub-acute cervicothoracic pain or radicular pain due to insufficient evidence, TENS is recommended as an adjunct intervention to an exercise program [67]. 

Two systematic reviews addressing shoulder conditions similarly concluded the uncertainty of TENS effectiveness for rotator cuff impingement [98] or adhesive capsulitis [99]. Overall low quality of evidence and heterogeneity of the comparators, intervention, and outcomes contributed to the uncertainty. These reports included the parameters but did not address adverse events. Variability in stimulation intensity and treatment duration relating to the overall dose of TENS along with comparators ranging from placebo, ultrasound, heat, exercise, extracorporeal shock wave, or a single injection of glucocorticoid steroid contribute to the uncertainty of TENS effectiveness. 

Two RCTs addressing muscle-related pain associated with temporomandibular disorders (TMD) were included in a systematic review and meta-analysis that included seven other interventions. The two trials included a total of 89 participants and measured pain on the 100 mm VAS. A pairwise meta-analysis favored TENS over the control (effect size = 1.80 [0.0–2.7], *p* = 0.0001) [96]. 

Twelve trials of chronic low back pain were included in a review only if compared to a negative control (sham, placebo, and medication) or active control (other form of electrotherapy) [97]. The authors concluded the other forms of electrotherapy were more effective than TENS in pain reduction in periods of follow up less than six weeks, with no significant difference in assessments completed greater than 6 weeks. Parameters for TENS, and the negative and active controls were not well-defined, and trial durations varied from one week to two years. An analysis of secondary outcomes of function found significant differences favoring TENS over negative control at greater than six weeks and no significant difference between TENS and an active control for function at either time point. 

Another systematic review and meta-analysis looked specifically at non-pharmaceutical interventions for pregnancy-related low back pain [60]. Thirteen RCTs were included in the systematic review. Only six RCTs qualified for inclusion in the meta-analysis, yielding 693 patients. The primary aim of the meta-analysis was to compare the effectiveness of interventions with typical care for low back pain during pregnancy. In the meta-analysis, the treatment group included different interventions compared with the control of typical care of pregnant women. Treatments included exercise, manipulation (OMT), kinesiotape, ear acupuncture, TENS, and progressive muscle relaxation exercises to music. Overall, the interventions deemed effective for a reduction in LBP during pregnancy [95% CI: (0.00–0.05)], were muscle relaxation exercises accompanied by music and TENS, with both of these being more effective than other interventions. The definitions of parameters and dosage were not reported in this meta-analysis for TENS or exercise. This systematic review concludes that TENS and progressive muscle relaxation exercise accompanied by music were found to be the most effective interventions. It is difficult to draw any conclusions about TENS due to the small sample size. 

### 7.6. Pelvic Health 

The use of TENS for women with primary dysmenorrhea and following C-section [61,100,101] are conditions that t have emerging evidence for TENS. Using the PEDro quality scale, two studies’ quality were assessed as excellent, with one each rated as good or fair in a systematic review evaluating four studies including 260 participants with primary dysmenorrhea [100]. The pooled results of the primary outcome of pain (SMD = 1.34; 95% CI = 0.505, 2.262; *p* = 0.002) measured by VAS, and the large effect size (ES = 1.384) indicated that TENS was more effective when compared with sham. Three of these studies indicated there were no adverse events with the fourth absent of this information [100]. Another systematic review regarding primary dysmenorrhea included six TENS trials (227 participants), with methodology scored as 4.8 on the PEDRO quality scale [101]. The comparators included sham, acupressure, placebo pill, and three studies with a pre-post test design with no control group. Favorable results were observed for LF and HF TENS, with HF determined to have the greatest effect for this population. A meta-analysis was not performed due to considerable heterogeneity of the trials. Using TENS with or without analgesia following C-section was one of eight different non-pharmacological interventions (37 studies) included in a systematic review [61]. The eight TENS studies including 386 participants were assessed as producing low-certainty evidence suggesting that TENS may have an effect on McGill pain scores at 1 and 24 h post time points, but not at 6 and 24 h time points. No safety or adverse events were reported due to a lack of certainty.

### 7.7. Cancer and Neurologic Conditions

Cancer pain proposes a challenge for research because of the longstanding perception that TENS is contraindicated for this population. A comprehensive review of biophysical interventions used for pain management [105] suggests that although the effect of TENS on malignant cells and metastasis is unknown and evidence is characterized as low, TENS remains a contraindication when applied directly over the site of cancers and a precaution when applied distant to the area. Importantly, TENS can be utilized when a person is cancer free for greater than 5 years and TENS may be used for patients in palliative care when the benefits of pain reduction outweigh risks [105].

In addition, TENS is often evaluated in reviews along with multiple nonpharmacological interventions. One such review included 37 studies, only one of which pertained to TENS and adult cancer pain [62]. This single systematic review was graded moderate for quality of evidence and presented with mixed recommendations for pain reduction in people with cancer pain [106].

A systematic review addressing neuropathic pain associated with cancer included 16 total trials, of which 6 utilized TENS as a self-administered intervention. Five of the six trials reported reduction in pain intensity in 173 participants. Both HF and LF were found to be effective, and in one trial TENS (41.6% reduction) reduced VAS pain intensity score when compared with pharmacological intervention alone [83]. 

Our 2014 review [52] included a discussion of emerging evidence of TENS effectiveness in people with painful diabetic peripheral neuropathy (DPN), complex regional pain syndrome (CRPS), and pain following spinal cord injury. For this current review, there were no additional meta-analyses or systematic reviews to add to this positive support for TENS use. 

Two systematic reviews addressing chronic pain in people with multiple sclerosis (MS) offered conflicting outcomes. Again, the reviews covered a host of non-pharmacological interventions. Amatya included eight trials, one evaluating TENS, and determined that though all groups (HF, LF, and P TENS) improved, none were significantly different from the others [84]. Positive outcomes for people with pain associated with MS were found in two trials comparing TENS with placebo in a different systematic review evaluating multiple interventions [86].

According to the Global Burden of Disease study 2016, migraine is ranked as the second most disabling disorder worldwide [107]. Two meta-analyses indicate support for TENS as an intervention; however, outcome measures are not pain intensity but instead focus on the number of days/months with migraine, medication intake, or number of pain-free hours in a day. When comparing active-TENS to sham-TENS in four trials with 276 participants, TENS was effective in reducing the number of headaches/months by 50% as compared with sham treatment, reduction in medication intake, and increased participant satisfaction [85]. The trials included both HF and LF TENS stimulation applied to peripheral nerve points, including the trigeminal, supraorbital (a branch of the trigeminal), supratrochlear, and occipital nerves. The authors summarized the quality of the evidence to be low. Seven of thirty-seven trials in a different systematic review pertained to TENS as a preventative or acute onset intervention for migraine [82]. Small to medium size effects were noted for supraorbital or occipital site electrode application using 60 Hz, 30 µs, and 16 mA stimulation for 20–30 min. The conclusions were positive (small effect size −0.494, 95% CI: −0.799 to −0.188) for the use of TENS to decrease the number of headaches per month and to decrease medication intake. However, there were limitations of small sample sizes and problematic effect size calculations, resulting in exclusion of five trials on the prevention of headache from the pooled meta-analysis [82].

There remains a paucity of evidence related to effectiveness of TENS to address phantom limb or stump pain. After selective searches of this topic in 2010 and 2015, there were no trials sufficient for inclusion in an analysis [63]. Thirty people with CRPS were included in an RCT, which suggested significant improvements in the TENS vs. the placebo group [108]. Because of the difficult nature of treatment for this population, additional RCTs would be of scientific/clinical value.

## 8. State of the Evidence for TENS, 2022

The majority of systematic reviews suggest that the effects of TENS are undecided. A confounder for those reviews demonstrating positive outcomes may be the risk of bias. Paley and colleagues [56] performed a comprehensive appraisal of the characteristics of over 169 systematic reviews on TENS and showed positive benefits in 69, no benefit in 13, and inconclusive in 87. Of 49 meta-analyses, only 3 pooled sufficient data (>500 participants) and all showed efficacy of TENS with lower pain in those with chronic musculoskeletal pain or labor pain, and lower analgesic consumption after surgery. In comparison to our 2014 review [52], there appears to be improvement in adverse events and parameter reporting. Importantly, stimulation intensity has been documented as critical to therapeutic success [1,26,72,109]. Examination of outcomes beyond resting pain, analgesic tolerance, and identification of TENS responders remain less studied areas of research. Our summary of the literature supports the conclusion that TENS may have efficacy for a variety of acute and chronic pain conditions, although the magnitude of the effect remains uncertain due to the low quality of existing literature.

## 9. Future Considerations

While TENS is simple to use and inexpensive, uncertainty over its effectiveness limits usage. The first clinical studies on TENS were published over 50 years ago, when effective parameters of stimulation (i.e., dose) were unclear and clinical trial design was in its infancy. Over the last two decades, a better understanding of the mechanisms underlying TENS efficacy led to a development of adequate dosing and understanding of its length of action. Clinical trial methodologies as well as the methodologies related to systematic reviews have simultaneously evolved. It is increasingly clear that to reduce risk of bias, RCTs need to be adequately powered, blinded, and randomized. Newer designs include pragmatic trials, enriched enrollment, “n of 1”, and target-based approaches may improve our understanding of TENS efficacy in a more real-world setting [110,111,112,113]. To begin to break down the uncertainty of evidence, those conducting future clinical trials and systematic reviews will need to consider a number of critical factors prior to deciding to implement these studies or analysis. Only when we begin to include these principles into clinical research on TENS can we begin to make informed decisions for those with chronic pain around the use of TENS. Thus, we suggest several key principles be applied to the design and implementation of any clinical trial or systematic review.

For future clinical trials, the following should be included in the design:Timing of outcome: pain during or immediately after TENS should be assessed. The greatest efficacy occurs during this time period [56,72,114].Intensity of stimulation: strong but comfortable intensity, or the highest tolerable intensity yet not painful. The greatest effects occur with stronger intensities [1,26,109].Sample size: Samples of ≥100 per group allows for adequate determination of effect size, better generalizability of results, and reductions in random error [64,115,116].Experimental design: a multisite or pragmatic design should be used to allow for better generalizability, larger sample sizes, and testing of the intervention in the setting and under the conditions in which it will be used, and other novel clinical trial approaches such as enriched enrollment, or “n of 1” designs, should be considered [110,111,112,113].Risk of bias: known risk of biases such as blinding, randomization, and use of adequate placebo should be controlled [115].Measure and report adverse events: serious and minor adverse events resulting from the study intervention should be reported. Few studies have collected this information for TENS. Those that have generally find few adverse events, and the adverse events that are found are minor [26].Responder analysis: parameters of the subjects and treatment that predict who will show the greatest response to TENS should be examined, which will allow us to better select subjects and to personalize treatments to the individual [2,117].

For future systematic reviews and meta-analyses, we recommend the following parameters be included in the inclusion/exclusion criteria and/or the reporting of data. 

Timing of outcome measurement: include data on pain (or primary outcome) during or immediately after treatment as this is the most effective time period for TENS [56,114].Intensity of stimulation: include studies with adequate stimulation parameters, particularly intensity. Strong but comfortable intensity, or highest tolerable intensity yet not painful are shown to produce the greatest effects [1,26,109].Sample size: perform meta-analyses only when pooled samples are of sufficient size to ensure generation of adequate effect sizes: of ≥500 per group. Consider not including RCTs with samples sizes of <50 per group. Allows adequate determination of effect size, better generalizability of results, reduction in heterogeneity, reduction in risk of bias, and reduction random error [64,115,116].Experimental design: Be cognizant of factors related to TENS efficacy in the design of the systematic review and meta-analysis. Include studies that use adequate dosing of TENS, adequate assessment of effects during or immediately after TENS, and repeated dosing of TENS. TENS parameters and assessment timing are critical to success of TENS and thus must be considered in the design of a systematic review and meta-analysis [3,64].Risk of bias: report on risk of biases such as blinding, randomization, and use of adequate placebo, and grade the evidence [64].Report adverse events: look for safety and efficacy data. While few studies have collected this information for TENS, it is imperative to weigh the risk relative to the benefit of an intervention [115].

## 10. Conclusions

In summary, while the literature on TENS is mixed, increasingly well-designed well-powered studies are showing efficacy when compared with placebo or no treatment. In many countries, TENS is available over the counter without a prescription and readily available as a self-management tool. Undoubtedly, TENS will continue to be used for pain control, with or without efficacy data. However, to provide information so that individuals with pain and the clinicians treating those with pain can make the most informed decisions, we suggest that resources for research should target larger, high-quality clinical trials and that systematic reviews and meta-analyses should focus only on areas with sufficiently strong clinical trials that will result in adequate sample size. These trials and systematic review should be cognizant of including adequate dose of TENS and adequate timing of outcome and should monitor the risks of bias. 

## Figures and Tables

**Table 1 medicina-58-01332-t001:** Review Summary Table 2014–2022. Ordered by Topics: Acute Pain; Cancer, Neurological; Chronic Pain, Fibromyalgia, Knee Osteoarthritis, Musculoskeletal, Pelvic Health.

Year	Topic	Author	Review Type	Studies (*n*)	Participants (*n*)	Summary	Ref
2021	Acute Pain	Davis	Systematic Review	1	72	This review was in a prehospital setting focused on paramedic pain management of femur fractures. One of the 19 articles utilized active TENS compared to sham TENS. Significant reduction in pain was noted with use TENS.	[80]
2019	Acute Pain	Binny	Systematic Review	3	192	The three studies included a comparison of active TENS to placebo TENS. Variable parameters were used among all three studies, with one occurring for one 30 min session during transport to the hospital, TENS before an exercise program over 4 weeks, and the third study reviewed TENS 2 x a week for 5 weeks. The risk of bias for these studies were rated as high.	[53]
2015	Acute Pain	Johnson	Review (Cochrane)	19	1346	Tentative evidence that TENS reduces pain intensity greater than placebo (no current) TENS as a stand-alone treatment for acute pain in adults. Authors included studies where a strong but comfortable intensity was utilized. The studies were rated as having a high risk of bias, inadequate sample size, and limited blinding. Treatment parameters were incomplete for replication of the studies.	[1]
2014	Acute Pain	Simpson	Systematic Review and Meta-Analysis	4	261	TENS produced a clinically significant reduction in severity acute of pain (mean VAS reduction) for patients with moderate-to-severe acute pain. TENS mean pain scores post-treatment were significantly lower than ‘sham’ TENS. TENS should be considered by emergency medical service providers when pharmacological pain management is restricted or unavailable.	[81]

2020	Cancer, Neurological	Moisset	Systematic Review and Meta-Analysis	38 7 TENS	745-acute 189 (TENS) 2846-prevention 456(TENS)	Supra-orbital transcutaneous electrical nerve stimulation (TENS), percutaneous electrical nerve stimulation (PENS), and high-frequency repetitive transcranial magnetic stimulation (rTMS) over the motor cortex (M1) are effective for migraine prevention. Two studies of moderate and very high quality that tested supra-orbital transcutaneous electrical nerve stimulation (TENS) for acute treatment were positive for their primary outcomes and most secondary outcomes. Variable quality of studies was noted.	[82]
2020	Cancer, Neurological	Ogle	Systematic Review	16	197	TENS was identified as a self-management strategy that may be helpful to patients experiencing peripheral neuralgia; however, this recommendation is based on low quality studies. Management by a clinician including adjunct interventions in the treatment of pain is warranted.	[83]
2018	Cancer, Neurological	Amatya	Systematic Review (Cochrane)	10	565	Review of non-pharmacological interventions for chronic pain in MS. Interventions reviewed included transcutaneous electrical nerve stimulation (TENS), psychotherapy (telephone self-management, hypnosis, and electroencephalogram (EEG) biofeedback), transcranial random noise stimulation (tRNS), transcranial direct stimulation (tDCS), hydrotherapy (Ai Chi), and reflexology. Result was a low level of evidence; variable pain measures and comparison groups.	[84]
2018	Cancer, Neurological	Tao	Systematic Review and Meta-Analysis	4	231	This study found a significant reduction in monthly headache days and medication intake for participants who received active TENS compared with sham TENS. The four studies included demonstrated lower-quality evidence limiting full endorsement of TENS. TENS may be of value for patients with or at risk for medication overuse.	[85]
2017	Cancer, Neurological	Gibson	Review of Reviews (Cochrane)	15	724	The review reported on active TENS compared with sham TENS. Eleven of the studies were rated as having a high level of bias, and many studies had a small sample size. For the pooled analysis of five studies, the evidence level was rated as low.	[54]
2015	Cancer, Neurological	Johnson	Systematic Review (Cochrane)	0	0	There were no RCTs meeting inclusion criteria to judge the effectiveness of TENS for phantom limb and stump pain. RCTs with rigor are required in order to make an assessment.	[63]
2014	Cancer, Neurological	Bao	Overview of Systematic Reviews	27	88	Review of complementary and alternative medicine for pain in adults with cancer. Results were inconsistent for massage therapy, transcutaneous electric nerve stimulation (TENS), and Viscum album L plus cancer treatment. However, the evidence levels for these interventions were low or moderate due to high risk of bias and/or small sample size of primary studies.	[62]
2014	Cancer, Neurological	Jawahar	Systematic Review	2	105	TENS was evaluated against placebo TENS in individuals with multiple sclerosis. Other physical therapy interventions were included in the review. Only TENS was identified as a promising non-pharmacological intervention for chronic pain. LF TENS demonstrated the greatest reduction in pain scores.	[86]

2021	Chronic Pain	Paley	Systematic Review and Meta-Analysis	169	Variable >500 (1); 200 to 499 (18); <200 (11) unclear (7)	A comprehensive appraisal of the characteristics of over 169 systematic reviews on TENS showed positive benefits in 69, no benefit in 13 and inconclusive in 87. Lower pain intensity was found during TENS compared with control for chronic musculoskeletal pain and labor pain, and lower analgesic consumption was found post-surgery during TENS use.	[56]
2019	Chronic Pain	Gibson	Systematic Review (Cochrane)	8	2895	This is a review of reviews. The reviews that were assessed found good methodology and low quality of evidence with small sample sizes. The summary was rated as uncertain for TENS compared with sham TENS, usual care/no treatment or with TENS combined with another active treatment compared with the active treatment alone. Heterogeneity in reviews was variable as well. Recommendations were made to improve future studies for TENS in individuals with chronic pain.	[87]
2018	Chronic Pain	Almeida	Systematic Review and Meta-Analysis	8	825	Review of the effect of TENS and IFC for acute pain and chronic pain. Transcutaneous electrical nerve stimulation and interferential current have similar effects on pain outcome. Overall, both TENS and IFC demonstrated pain reduction and improved function.	[55]

2018	Fibromyalgia	Honda	Systematic Review and Meta-Analysis	11	498	Review of physical agent modalities of low-level laser therapy (LLLT), thermal therapy, electromagnetic field therapy, and transcutaneous electrical nerve stimulation (TENS). Electromagnetic field therapy was associated with significantly reduced VAS score and FIQ score. Active TENS compared with control group show significantly reduced VAS scores.	[88]
2017	Fibromyalgia	Johnson	Systematic Review (Cochrane)	8	315	Review of eight trials (RCTs and quasi RCT) with a high risk of bias in seven of the eight studies. Focus was on reporting of pain relief of ≥ 30%, ≥ 50% and patient global impression of change (PGIC). Active TENS was effective at relieving pain associated with fibromyalgia, but the studies had very small sample sizes and were underpowered, resulting in uncertain evidence.	[57]
2017	Fibromyalgia	Salazar	Systematic Review and Meta-Analysis	9	301	This review found electrical stimulation as an adjunct treatment option providing improvement in pain relief for patients with FM. Low-quality evidence for the effectiveness of electrical stimulation for pain reduction in patients with fibromyalgia. A variety of TENS parameters and frequency of TENS application noted. Moderate-quality evidence for the effectiveness of electroacupuncture combined or not combined with other types of treatment.	[89]

2021	Knee Osteoarthritis	Shi	Systematic Review and Meta-Analysis	4	116	This meta-analysis combined studies comparing high-frequency TENS to a placebo or no treatment in individuals with knee OA. High-frequency TENS reduced pain more than the control intervention.	[90]
2017	Knee Osteoarthritis	Li	Systematic Review and Meta-Analysis	5	472	This review of RCTs was focused on pain and opioid consumption following TKA at 12 h, 24 h, and 48 h. Secondary outcomes included length of stay, nausea, and vomiting. The application of TENS demonstrated greater reduction in VAS scores and opioid consumption at 12, 24, and 48 h after TKA compared with placebo TENS. In addition, there was a decreased risk of nausea and vomiting in experimental groups compared with control groups.	[91]
2017	Knee Osteoarthritis	Zhu	Systematic Review and Meta-Analysis	6	529	Review of RCTs examining TENS as an adjunctive therapy following TKA compared with a control intervention. Active TENS reduced pain and total postoperative morphine dose over a 24 h period following TKA compared with the control group. At 2 weeks post-surgery, no difference was noted between the TENS and control groups.	[92]
2016	Knee Osteoarthritis	Chen	Systematic Review and Meta-Analysis	12	792	Active TENS compared with control significantly reduced pain. Follow-up time points ranged from 0.5 to 6 months.	[93]
2016	Knee Osteoarthritis	Cherian	Systematic Review	7	70	Seven studies with use of active TENS showed pain reduction from pre-treatment to post treatment. Follow-up times mean was 8 weeks.	[94]
2015	Knee Osteoarthritis	Zeng	Systematic Review and Meta-Analysis	27	1249	Review of pain relief in individuals with knee OA for six types of electrical stimulation: high-frequency transcutaneous electrical nerve stimulation (h-TENS), low-frequency transcutaneous electrical nerve stimulation (l-TENS), neuromuscular electrical stimulation (NMES), interferential current (IFC), pulsed electrical stimulation (PES), and noninvasive interactive neurostimulation (NIN). Effectiveness was based on change in pain intensity and change in pain score. IFC is significantly effective treatment in terms of both pain intensity and change pain score at last follow-up time point when comparing with the control group. HF TENS decreased the pain score compared with control groups but not LF TENS.	[95]

2022	Musculoskeletal	Ferrillo	Systematic Re- view and Meta- Analysis	2	89	This review of muscle-related pain included eight interventions. The two trials assessing TENS efficacy used pain as the primary outcome. One trial was a single 50 min session, and the other was TENS for 1 h/day for 10 weeks. Pairwise meta- analysis demonstrated that TENS was favored over control.	[96]
2021	Musculoskeletal	Koukoulithras	Systematic Review and Meta-Analysis	6	20	This is a review of the effectiveness on non-pharmacological interventions for individuals with low back pain and pregnancy-related low back pain. A variety of interventions were reviewed: exercise, manipulation, ear acupuncture, Kinesio tape, transcutaneous electrical nerve stimulation (TENS), and neuroemotional technique. TENS and progressive muscle relaxation exercise with music were more effective than the other interventions.	[60]
2019	Musculoskeletal	Martimbianco	Systematic Review (Cochrane)	7	651	This review focused on the use of active TENS compared with sham TENS in individuals with chronic neck pain. Variability noted in the heterogeneity of the studies. This review found very low certainty of evidence for a difference between TENS compared with sham TENS on reducing neck pain.	[59]
2018	Musculoskeletal	Wu	Systematic Review and Meta-Analysis	12	700	Review of TENS in treatment for individuals with chronic back pain. TENS was compared with a control, sham, placebo, and other types of nerve stimulation therapies (NSTs) including electroacupuncture, Percutaneous electrical nerve stimulation (PENS) and percutaneous neuromodulation therapy (PNT). TENS was more effective than the control group in improving functional disability only in patients with follow-up of < 6 weeks. TENS was similar to the control treatment for providing pain relief, but other nerve stimulation therapies were more effective.	[97]
2016	Musculoskeletal	Page	Systematic Review (Cochrane)	47	2388	This review focused on electrotherapy modalities for individuals with rotator cuff disease. Interventions included therapeutic ultrasound, low-level laser therapy (LLLT), transcutaneous electrical nerve stimulation (TENS), and pulsed electromagnetic field therapy (PEMF). One TENS study compared TENS with placebo in 20 participants. Another trial was not significantly different for a decrease in pain comparing TENS plus hot pack vs. hot pack. Results are uncertain when examining TENS effectiveness compared with glucocorticoid injection with respect to pain, function, global treatment success, and active range of motion due to very low-quality evidence from a single trial.	[98]
2014	Musculoskeletal	Page	Systematic Review (Cochrane)	19	1249	A review of multiple modalities for participants with shoulder adhesive capsulitis. The review included four studies with TENS in combination with other treatment strategies but did not compare active TENS alone to placebo or no treatment. The review found low- or very low-quality evidence and reported uncertainty whether any of the modalities, including TENS in combination with other modalities, were effective as adjuncts to exercise.	[99]

2022	Pelvic Health	Arik	Systematic Review and Meta-Analysis	4	260	Review to evaluate the effectiveness of TENS in the treatment of pain in women with primary dysmenorrhea with positive results with the comparison of active TENS and sham.	[100]
2020	Pelvic Health	Zimpel	Systematic Review (Cochrane)	4	278	Review of complementary and alternative therapies (CAM) for individuals with post-caesarean pain. CAM studies included the interventions of acupuncture or acupressure, aromatherapy, electromagnetic therapy, massage, music therapy, relaxation, and TENS. There was a great deal of heterogeneity among the studies. Quality of evidence varied from low to moderate. TENS (versus no treatment) may reduce pain at one-hour TENS plus analgesia (versus placebo plus analgesia) may reduce pain compared with placebo plus analgesia at one hour and at 24 h. TENS plus analgesia (versus placebo plus analgesia) may reduce heart rate and respiratory rate.	[61]
2016	Pelvic Health	Igwea	Systematic Review	6	461	Review of TENS and heat therapy for pain reduction and improvement in quality of life for women with primary dysmenorrhea. TENS and heat therapy both show evidence of pain reduction, but no study included quality of life as an outcome. TENS types varied between strong low-rate acupuncture-like TENS, sham TENS, and HF TENS.	[101]

**Table 2 medicina-58-01332-t002:** Review Outcome Measures, Adverse Events, TENS Ratings, and TENS Recommendations.

Year	Topic	Author	Pain Outcome Measures	Movement Evoked Pain	Adverse Event Reported	Rating Positive (+) Negative (−) Equivalent (=) Undecided (u)	TENS Recommendation	Ref
2021	Acute Pain	Davis	Pain scores, not specified	No	Reported no adverse events	(+)	Promising results for patients with hip fractures in the prehospital setting and would benefit from further studies	[80]
2019	Acute Pain	Binny	VAS, NRS	No	Limited data with 2 studies reporting no AE’s.	(u)	Recommended further studies	[53]
2015	Acute Pain	Johnson	VAS, NRS < VRS, MPQ	No	Yes	(+)	While TENS use for acute pain in adults remains a matter of debate, it compares favorably with many alternatives because it is inexpensive, self-administered, safe, and readily available to patients	[1]
2014	Acute Pain	Simpson	VAS	N/A	No safety risks identified	(+)	Emergency medical services should consider TENS when pharmacological pain management is unavailable or restricted.	[81]

2020	Cancer, Neurological	Moisset	HA days per month	No	No	(+)	TENS may be effective for acute migraine HA; larger well-conducted studies are necessary to confirm efficacy	[82]
2020	Cancer, Neurological	Ogle	VAS, NRS, SF-MPQ	No	No	(+)	TENS as a self-management strategy monitored by a clinician may be beneficial in reduction in peripheral neuropathy pain	[83]
2018	Cancer, Neurological	Tao	HA days per month	No	Yes	(+)	TENS may be an effective alternative to reduce monthly HA days. Well-designed RCTs are necessary to confirm and update findings.	[85]
2018	Cancer, Neurological	Amatya	VAS, BPI, NRS	No	No	(u)	Further studies with larger sample sizes	[84]
2017	Cancer, Neurological	Gibson	VAS	No	Three studies reported AE. AEs included skin irritation	(u)	Improve the quality of design of TENS studies	[54]
2015	Cancer, Neurological	Johnson	No Articles to review	No Articles to review	No Articles to review	(u)	No articles to review	[63]
2014	Cancer, Neurological	Bao	VAS, NRS	No	No	(+)	TENS might have beneficial for pain reduction in bone cancer; small sample sizes	[62]
2014	Cancer, Neurological	Jawahar	VAS, McGill Pain Questionnaire	No	No	(+)	TENS may be effective in reducing central neuropathic pain in MS. Recommendations were made for more rigorous design and reporting is needed to determine TENS effectiveness for individuals with MS.	[86]

2021	Chronic Pain	Paley	VAS, Estimated Effect	No	Yes	(+)	Multiple reviews with multiple conditions. TENS has a tendency toward benefit in 16/169 reviews, no benefit in 13/169 reviews and inconclusive in 87/168 reviews. Inconsistency in data limiting recommendations. Recommendations made to improve data collection in future studies.	[56]
2019	Chronic Pain	Gibson	VAS, NRS	No	Three studies reported AE. AEs for the studies reported were primarily skin irritation.	(u)	Recommendations for future studies focused on the comparison groups, timing of pain ratings, data for parameters for reproducibility with adequate intensity and larger sample sizes.	[87]
2018	Chronic Pain	Almeida	VAS	No	No	(+)	TENS and IFC had positive effects on pain level and function	[55]

2018	Fibromyalgia	Honda	VAS	No	No	(+)	Further studies needed	[88]
2017	Fibromyalgia	Johnson	VAS, NRS, Pain relief ≥ 30%	Yes/No	Withdrawal due to increased pain, no reasons given for some of the studies	(u)	Further high-quality studies are needed.	[57]
2017	Fibromyalgia	Salazar	VAS	No	No	(u)	Biases noted with the studies and further research with high quality studies.	[89]

2021	Knee Osteoarthritis	Shi	VAS	No	No	(+)	None	[90]
2017	Knee Osteoarthritis	Li	VAS	No	No	(+)	Further high-quality studies are needed.	[91]
2017	Knee Osteoarthritis	Zhu	VAS in 24 h post-surgery	No	No	(+)	Further studies needed for duration and intensity of TENS.	[92]
2016	Knee Osteoarthritis	Chen	VAS	No	No	(+)	Further RCT with studies with larger sample sizes and longer follow up time frame.	[93]
2016	Knee Osteoarthritis	Cherian	VAS	No	No	(+)	Further long-term studies are needed.	[94]
2015	Knee Osteoarthritis	Zeng	VAS, WOMAC, Present Pain Intensity	No	Reported in 7 of 27 studies with no AEs related to TENS	(−)	None	[95]

2022	Musculoskeletal	Ferrillo	VAS	No	No	(+)	TENS may decrease pain after as single 50 min session and over 25 weeks for 10 weeks	[96]
2021	Musculoskeletal	Koukoulithras	VAS	No	No	(+)	May be helpful, further studies needed.	[60]
2019	Musculoskeletal	Martimbianco	VAS	No	No	(u)	Further high-quality studies are needed.	[59]
2018	Musculoskeletal	Wu	VAS, NRS, McGill Pain Questionnaire, Borg Verbal rating scale (BPS)	No	No	(+), (=)	TENS was found to improve function disability after within 6 weeks of the treatment.	[97]
2016	Musculoskeletal	Page	VAS	No	No	(u)	Recommendations for TENS were uncertain if TENS is more or less effective than glucocorticoid injection with respect to pain, function, global treatment success and active range of motion because of the very low-quality evidence from a single trial.	[98]
2014	Musculoskeletal	Page	VAS, Pain relief > 30%	No	No	(u)	Further studies needed.	[99]

2022	Pelvic Health	Arik	VAS, NRS	No	AE reporting in 3 of 4 studies; No AE reported in these 3 studies	(+)	TENS is safe and well-tolerated and has shown evidence of pain reduction in primary dysmenorrhea	[100]
2020	Pelvic Health	Zimpel	VAS	No	No	(+), (u)	TENS plus analgesia may be of benefit in the first 24 h.	[61]
2016	Pelvic Health	Igwea	VAS, NRS, McGill Pain Questionnaire	No	No	(+)	Additional rigorous high-quality trials are still needed to make conclusive recommendations.	[101]

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
