# Peer review of "Using TENS for Pain Control: Update on the State of the Evidence"

_medicina, 2022, doi:10.3390/medicina58101332_

Round 1
Reviewer 1 Report
The authors made a review of the literature regarding the use of TENS for pain control. it is a very nice narrative demonstrating the effectiveness of TENS of pain however they give attention for the need of better clinical trials for better understanding of the use of TENS. The authors revised the mechanisms underlying TENS the evidences for acute and chronic pain. it is a very nice and well written review of the literature
Author Response
Thank you for your review of our manuscript. We have run spell check to ensure proper spelling and grammar suitable for publication in the journal.
Reviewer 2 Report
The paper "Using TENS for pain control: Update on the state of the evidence” by
Carol GT Vance et al. presents a fresh overview of Transcutaneous Electrical Nerve Stimulation (TENS) as a non-pharmacological intervention treatment method, with a focus on the quality of the existing literature and evidence to support a wider use of TENS in the clinical practice.
The authors conclusions are more of recommendations drawn from the critical review of cited references coupled the extensive experience of their own research group.
As such, the authors offer some valuable suggestions on the how to plan the future high quality clinical trials (to include adequate TENS dose, adequate timing of outcome and monitor risks of bias) while for the systematic reviews and
meta-analyses recommend a focus on areas with sufficiently strong clinical trials that will result in adequate sample size.
Comments & Questions for the authors:
Minor
1. Some review of the English spelling is required. (e.g. “This is turn activates […]” page 2/23.
Author Response
Thank you for your review of our manuscript. We have run a spell check on the document for proper grammar and spelling acceptable for publication. We intend to leave the sentence you mentioned on p.2 with a change from “is” to “in” to read as -"This in turn activates descending inhibitory systems that reduce hyperalgesia." We feel the sentence reflects our intended thought for the reader. The sentence prior to it suggests a first thought (activation of large fibers) and this sentence indicates the results of the large fiber activation (activation of descending inhibitory systems and resultant hyperalgesia). If we are missing the point of the problem with this sentence, please advise us further.
Reviewer 3 Report
Dear Authors,
The aim of this study was to review the fundamental mechanisms of TENS as well as an up-to-date critique of current clinical research for TENS.
The study is of scientific interest and in line with the aims of the journal.
The author guidelines have been respected and the work is well written.
Introduction
· “Transcutaneous Electrical Nerve Stimulation (TENS) is a non-pharmacological intervention used in the treatment of acute and chronic pain conditions.” Please add references “Wu LC, Weng PW, Chen CH, Huang YY, Tsuang YH, Chiang CJ. Literature Review and Meta-Analysis of Transcutaneous Electrical Nerve Stimulation in Treating Chronic Back Pain. Reg Anesth Pain Med. 2018 May;43(4):425-433. doi: 10.1097/AAP.0000000000000740. PMID: 29394211; PMCID: PMC5916478.”; “Ferrillo et al. Efficacy of rehabilitation on reducing pain in muscle-related temporomandibular disorders: A systematic review and meta-analysis of randomized controlled trials. J Back Musculoskelet Rehabil. 2022 Feb 18. doi: 10.3233/BMR-210236.”; “Martimbianco ALC, Porfírio GJ, Pacheco RL, Torloni MR, Riera R. Transcutaneous electrical nerve stimulation (TENS) for chronic neck pain. Cochrane Database Syst Rev. 2019 Dec 12;12(12):CD011927. doi: 10.1002/14651858.CD011927.pub2.”.
· “TENS units are safe, inexpensive, over-the counter devices that deliver pulsed alternating current applied through electrodes placed on the skin”, please add references.
· Please modify references according to authors guidelines. (https://www.mdpi.com/journal/medicina/instructions)
1. Author 1, A.B.; Author 2, C.D. Title of the article. Abbreviated Journal Name Year, Volume, page range.
Author Response
Thank you for your review of our manuscript. We have run a spell check on the document for proper grammar and spelling acceptable for publication.
We have added two references to the second sentence in the introduction, “TENS units are safe, inexpensive, over-the counter devices that deliver pulsed alternating current applied through electrodes placed on the skin” as suggested. The manuscripts Johnson MI, Paley CA, Howe TE, Sluka KA. Transcutaneous electrical nerve stimulation for acute pain. The Cochrane database of systematic reviews. 2015;2015:Cd006142., and Vance CGT, Zimmerman MB, Dailey DL, Rakel BA, Geasland KM, Chimenti RL, et al. Reduction in movement-evoked pain and fatigue during initial 30-minute transcutaneous electrical nerve stimulation treatment predicts transcutaneous electrical nerve stimulation responders in women with fibromyalgia. Pain. 2021;162:1545-55. Both manuscripts depict the utilization of TENS in acute (Johnson et. al) and chronic (Vance et. al) pain populations; and both include discussion about TENS safety.
We respectfully decline to add any references to the opening sentence in the introduction, “Transcutaneous Electrical Nerve Stimulation (TENS) is a non-pharmacological intervention used in the treatment of acute and chronic pain conditions.” This statement is made routinely in the introduction of many manuscripts about TENS and is considered common knowledge to the degree that we would have a difficult time selecting only a few references to include for this opening statement.
Two of the references suggested (Wu et al 2018 and Martimbianco et al 2019) are included in the manuscript in the musculoskeletal section for review by the reader. Thank you for suggesting inclusion of the third reference (“Ferrillo et al. Efficacy of rehabilitation on reducing pain in muscle-related temporomandibular disorders: A systematic review and meta-analysis of randomized controlled trials. J Back Musculoskelet Rehabil. 2022 Feb 18. doi: 10.3233/BMR-210236). It has been included in the text and tables to improve our review.
Thank you for your suggestion for modification of the references according to authors guidelines. We have completed this edit.
Round 2
Reviewer 3 Report
Dear Authors,
you modified the text according to the suggestions.
In my opinion, it is suitable for publication.